# Antisolvent Engineering to Enhance Photovoltaic Performance of Methylammonium Bismuth Iodide Solar Cells

**DOI:** 10.3390/nano13010059

**Published:** 2022-12-23

**Authors:** Ming-Chung Wu, Ching-Mei Ho, Kai-Chi Hsiao, Shih-Hsuan Chen, Yin-Hsuan Chang, Meng-Huan Jao

**Affiliations:** 1Department of Chemical and Materials Engineering, Chang Gung University, Taoyuan 33302, Taiwan; 2Green Technology Research Center, Chang Gung University, Taoyuan 33302, Taiwan; 3Division of Neonatology, Department of Pediatrics, Chang Gung Memorial Hospital at Linkou, Taoyuan 33305, Taiwan

**Keywords:** methylammonium bismuth iodide, solar cell, anti-solvent, lead-free

## Abstract

High absorption ability and direct bandgap makes lead-based perovskite to acquire high photovoltaic performance. However, lead content in perovskite becomes a double-blade for counterbalancing photovoltaic performance and sustainability. Herein, we develop a methylammonium bismuth iodide (MBI), a perovskite-derivative, to serve as a lead-free light absorber layer. Owing to the short carrier diffusion length of MBI, its film quality is a predominant factor to photovoltaic performance. Several candidates of non-polar solvent are discussed in aspect of their dipole moment and boiling point to reveal the effects of anti-solvent assisted crystallization. Through anti-solvent engineering of toluene, the morphology, crystallinity, and element distribution of MBI films are improved compared with those without toluene treatment. The improved morphology and crystallinity of MBI films promote photovoltaic performance over 3.2 times compared with the one without toluene treatment. The photovoltaic device can achieve 0.26% with minor hysteresis effect, whose hysteresis index reduces from 0.374 to 0.169. This study guides a feasible path for developing MBI photovoltaics.

## 1. Introduction

Metal-halide perovskites have drawn significant attention as one of the most promising next-generation photovoltaic materials due to their adjustable composition, direct bandgap, long carrier diffusion length, and outstanding optoelectronic properties [1,2,3,4,5,6,7]. Since 2009, the power conversion efficiency (PCE) has increased from 3.8% to 25.7% in 2022 [8,9]. Despite these excellent material merits, problems associated with the toxicity of Pb-based perovskite solar cells must be solved before their commercial deployment [10,11]. For the perovskite active layer, the crystal-structure transition is seen as an important factor that could affect stability. In this point of view, based on Goldschmidt tolerance factors and octahedral factors, multiple potential alternatives were selected, such as tin [12,13], strontium [14,15], barium [16,17], and bismuth [18,19]. Tin-based perovskite material has gained immense consideration due to its comparable optoelectronic properties to lead-based perovskite. However, the oxidation of Sn^2+^ to Sn^4+^ in perovskite films shows serious stability issues, making the device conversion efficiency being way behind the expectation [20,21,22].

Recent studies have demonstrated promising methylammonium bismuth iodide perovskite-derivative materials ((CH_3_NH_3_)_3_Bi_2_I_9_, MBI) as a lead-free light absorber due to their nontoxicity and stability [23,24,25]. Jain et al. fabricated lead-free MBI-based solar cells with a non-toxic solvent system. The strongly quenched PL observed in MBI/poly(3-hexylthiophene-2,5-diyl) (P3HT) films shows efficient charge separation at the MBI/P3HT interface. The device can maintain its initial performance with minor decay after 300 h of light soaking test under continuous 1 sun illumination without encapsulation [26]. Kong et al. fabricated MBI films through a two-step soaking-assisted sequential solution (2-S) method with compact TiO_2_ (cp-TiO_2_) and mesoporous TiO_2_ (mp-TiO_2_) layer. Due to the difference crystallization rates for the one-step and 2-S methods, the MBI film transformed from the BiI_3_ thin film (2-S) showed higher homogeneity and coverage on the TiO_2_ layer [27]. Hu et al. demonstrated the fabrication of MBI layers by the chemical vapor deposition process, enabling scale-up manufacturing. However, the weak penetration of MBI into mp-TiO_2_ limited the carrier transport and device performance [28]. The major issue causing the limited photovoltaic performance of MBI-based devices is short diffusion length of their carriers, which is one to two order lower than conventional perovskite materials [29,30,31,32]. As a result, improving morphology and crystallinity is a path to mitigate the energy loss from defect-assisted recombination. The improvement of MBI morphology by controlling the nucleation and growth of MBI crystals is essential to enhance the film quality and corresponding photovoltaic performance.

Anti-solvent-assisted crystallization strategy is a common method in perovskite photovoltaics to facilitate the removal of the residual solvent and delicately control the nucleation condition and crystallization rate of the perovskite film [33,34,35]. The selection of anti-solvent and deposition routes has great impact on the nucleation speed during the perovskite fabrication. Taylor et al. demonstrated 14 different types of anti-solvents for the Cs_0.05_(MA_0.17_FA_0.83_)_0.95_Pb(I_0.9_Br_0.1_)_3_ solar cells. The appropriate anti-solvents can be mainly categorized into two types, which are miscible and immiscible with host solvents. The anti-solvents with different polarities show diverse interaction between anti-solvent and host solvent. Moreover, the miscibility of anti-solvent and host solvent determined the dripping timing and rate of crystallization [36]. The antisolvent with ability to dissolve the organic precursor components and miscibility with the host solvents has the largest process window for fabricating high-quality perovskite layers. However, fewer studies focused on anti-solvent engineering for lead-free perovskite or perovskite-derivative materials.

In this study, we explore four anti-solvents, including two miscible anti-solvents with host solvents: chlorobenzene and di-chlorobenzene and two immiscible anti-solvents with host solvents: diethyl ether and toluene, to discuss their effects on crystallization of methylammonium bismuth iodide films. Among them, toluene, as the anti-solvent, contributed to the high-quality methyl bismuth iodide film due to its low dipole moment and infiltration ability. The compacted surface morphology and high crystallinity of MBI from toluene-assisted crystallization directly reflects on the PV performance and also mitigates hysteresis phenomena. The PCE of the toluene-treated device can be achieved to 0.26% with a hysteresis index of 0.169. Our results represent a fundamental understanding of the role of anti-solvent in lead-free methyl bismuth iodide film formation.

## 2. Materials and Methods

### 2.1. Materials and Preparation

All chemicals in this study are used as received without further purification process. In terms of an electron transport layer in devices, it includes two components, compacted TiO_2_ layers, and mesoporous TiO_2_ layers. The precursor solution for the compacted TiO_2_ layer is prepared by 2.0 mL of titanium diisopropoxide bis(acetylacetonate) ((CH_3_)_2_CHO_2_Ti(C_5_H_7_O_2_)_2_, 75 wt%, Sigma-Aldrich, St. Louis, MO, USA) and 78.0 mL of ethanol (CH_3_CH_2_OH, >99.8%, Sigma-Aldrich, St. Louis, MO, USA). For mesoporous TiO_2_, the TiO_2_ paste is followed by our previous study [37]. Briefly, titanium dioxide nanocrystal is first prepared by the sol-gel method with 25.0 g of titanium isopropoxide (Ti(OCH(CH_3_)_2_)_4_, >97%, Sigma-Aldrich), 10.0 mL of 2-propanol ((CH_3_)_2_CHOH, IPA, >99.8%, STAREK), and 180 mL of acetic acid (CH_3_COOH, >99.7%, Sigma-Aldrich) with a concentration of 3.5 M. For further ripening of TiO_2_ nanocrystal, the translucent solution is further transferred into a Teflon-lined autoclave and is heated at 170 °C for 6 h. The obtained TiO_2_ nanocrystals are dispersed into diluted agents of α-terpineol (C_10_H_18_O, 90%, Merck, Rahway, NJ, USA) and ethyl cellulose (ethoxyl content 48%, 22 cps, Acros, Geel, Belgium) to obtain a meso-TiO_2_ paste with 23.0 wt% of TiO_2_ nanocrystals. For precursor of the light active layer, MBI, equal stoichiometric of methylammonium iodide (MAI, >99%, FrontMaterials Co. ltd., Taoyuan, Taiwan) and bismuth iodide (BiI_3_, 99.998%, Alfa Aesar, Haverhill, MA, USA) are weighted to the desired ratio. The precursors are dissolved into a mixed solvent system of dimethyl sulfoxide (DMSO, 99.7%, Acros) and dimethylformamide (DMF, 99.8%, Acros) with a volume ratio of 13 to 1 to obtain a solution with a concentration of 0.9 M. For the hole transport layer, 80 mg of spiro-OMeTAD and 25.8 μL of 4-tert-butylpyridine (tBP, >98%, Acros) are first dissolved into 1 mL of chlorobenzene (CB, 99.8%, Acros). The dopant of lithium-bis-(trifluoromethanesulfonyl)imide (Li-TFSI, 99.95%, Sigma-Aldrich) solution was first prepared by dissolving 104 mg of Li-TFSI into 200 μL of acetonitrile (ACN, 99.5%, Acros) and the dopant solution is added into the previous solution with a volume of 17.5 μL with continuous stirring to complete the preparation of HTM solution.

### 2.2. Device Fabrication

In this study, all device architectures followed the n-i-p mesoporous structure. To remove adsorbed contaminations, fluoride-doped tin oxide coated glass (FTO glasses, 7 Ω, FrontMaterials Co., ltd., Taoyuan, Taiwan) are sequentially washed with deionized water, and organic solvents, including acetone, and isopropanol. A compact TiO_2_ layer is deposited onto the conductive side of an FTO glass with the as-prepared titanium diisopropoxide bis(acetylacetonate) solution through spray pyrolysis. The as-prepared meso-TiO_2_ paste is thereafter screen-printed onto the FTO substrate with a compact TiO_2_ layer. The substrates with as-deposited meso-TiO_2_ and compacted TiO_2_ layer are calcined at 500 °C for 30 min and cooled down to room temperature. The light absorber layer of MBI is deposited onto the substrate with anti-solvent-assisted deposition. Briefly, the spin-coating process is carried out with two-step spin rate, 1000 rpm for 10 s, and 5000 rpm for 20 s. The anti-solvent of toluene is dripped onto the wet-film of MBI at the timing of 3 s before the coating process is finished. The as-prepared MBI films are placed onto a hot plate with a temperature of 100 °C for 30 min to increase their crystallinity and to ensure no solvent remaining in them. After the MBI deposition, a hole-transporting layer of sprio-OMeTAD is deposited on the MBI layer by spin-coating at 2500 rpm for 30 s. Twelve silver electrodes with area of 0.09 cm^2^ confining by a metal mask are thermal evaporated onto the HTM layer to complete the fabrication of devices.

### 2.3. Characterization

UV-vis spectrometer (V-730, Jasco, Tokyo, Japan) is used to measure the optical property of MBI films deposited on quartz substrates. X-ray diffractometer (D2 phaser, Bruker, Karlsruhe, Germany) is applied to characterize the crystal structure of MBI films. A field-emission SEM (FE-SEM, su8010, HITACHI, Tokyo, Japan) equipped with energy-dispersive X-ray spectroscopy (EDS) (Xflash Detector 5030, Bruker, Yokohama, Japan) is used to observe the surface morphology and element distribution of MBI films. For photovoltaic performance, the solar simulator source (YAMASHITA DENSO, YCSS-50, class AAA, Tokyo, Japan) is used to analyze the photovoltaic performance of devices. The photocurrent density (J)-voltage (V) curve of MBI devices are all measured in air under A.M. 1.5 irradiation (100 mW/cm^2^) and acquired by the Keithley 2400 source meter.

## 3. Results

In order to realize the ambipolar properties of methylammonium bismuth iodide ((CH_3_NH_3_)_3_Bi_2_I_9_, MBI) layers, the precursor of methylammonium iodide (MAI) and bismuth iodide (BiI_3_) should be controlled to a certain stoichiometry. The anti-solvent of toluene is preliminarily selected to assist the crystallization of MBI films. Based on different solubility of precursors of BiI_3_ and MAI in the solvent system, the ratio of the reactants should be carefully controlled to obtain the desired MBI materials. The relationship between ratio of bismuth/methylammonium and PCE from devices with such light absorber layers is shown in Figure 1. Based on the photovoltaic parameters, the ratio of 1 on 1 shows superior PV performance to the others, especially in terms of open-circuit voltage (*V_OC_*) and short-circuit current (*J_SC_*). That can be ascribed to different precursor solubility in the solvent system as anti-solvent dripping. As a result, the equal stoichiometric of reactants can obtain the desired MBI with mere impurity (discussed later in the XRD patterns). The balance between constructed cations and anions in MBI influences the energy level as elements bond with each other. Generally, 6 p orbital of bismuth predominates the conduction band, whereas the 5 p orbital of iodide from bismuth iodide or methylammonium iodide dominates the valance band of MBI materials [38]. Therefore, the n-type or p-type nature of MBI is highly related to the constructed ratio of bismuth and iodide in MBI materials. In addition to its ambipolar nature, the diffusion threshold of MBI is another factor influencing effective carrier transportation. By tuning the concentration of precursor, the thickness of MBI can be well controlled. The carrier diffusion and film thickness reach a balance as the precursor concentration of MBI is at 0.9 M as shown in Figure 2. That makes the devices with MBI from 0.9 M precursor solution perform the highest PCE compared with the others. In addition, the annealing temperature of MBI play an influential role in their crystallization. Moreover, it can promote the crystallinity of MBI layers. The high crystallinity of MBI film is speculated to have fewer macro defects such as grain boundaries. That facilitates the carrier diffusion with minor decay, namely defect-assisted recombination. However, the high annealing temperature causes the rapid crystallization of MBI crystals that results in high surface roughness and thermal degradation of MBI precursors. Therefore, the device with MBI film annealed at 100 °C shows the highest PV performance as shown in Figure 3. Based on the preliminary results, the following discussion followed the optimized process with the stoichiometry of cation at 1 on 1, precursor concentration at 0.9 M, and annealing temperature at 100 °C.

Anti-solvent-assisted crystallization is widely used to deposit a uniform perovskite films with high crystallinity. With the assistance of poor solvent, the nucleation site can be generated as the anti-solvent dripping onto a wet film during film deposition. The low solubility of anti-solvent to precursor forces precursor to precipitate and act as nucleation sites in a wet film. A series of solvents, including miscible and immiscible with host solvent system, with low polarity are selected as anti-solvent to help the crystallization step of MBI films as shown in Table 1. The results reveal that the devices with anti-solvent treated MBI films show high reproducibility compared with devices without anti-solvent treated MBI film. With the anti-solvent dripping, the PV performances obviously increase compared to those without anti-solvent dripping. The photovoltaic performance including open-circuit voltage (*V_OC_*), short-circuit current (*J_SC_*), and fill factor (FF) from devices without anti-solvent dripping are inferior to those with anti-solvent treatment. The relationship between miscibility of anti-solvents and the host solvent obscures to the PV performance. As focusing on the difference of boiling point, diethyl ester (DEE), exhibiting the lowest boiling point, and 1, 2-dichlorobenzene (DCB), having the highest boiling point, show inferior PV performance in the series. That can be ascribed to the infiltration ability of solvents as they are dripped onto the wet films of MBI during spin-coating process. The moderate boiling point of anti-solvent helps to extract the precursor solvent of dimethyl sulfoxide (DMSO) and gamma-butyrolactone (GBL). Moreover, the dipole moment of anti-solvent is curial for creating a metastable condition instead of supersaturation condition [39]. As a result, the devices with toluene-treated MBI films perform the highest photovoltaic performance in the series. The corresponding champion devices are shown in Figure 4.

The giant difference in PV performance between MBI film with and without toluene treatment inspires us to investigate the effect of toluene on MBI films. The appearances of MBI films without and with toluene treatment are shown in Figure 5a,b. After toluene treatment, the film is darker than the one without toluene treatment. The corresponding UV-Vis spectrum is also shown in Figure 5c. The absorption of toluene-treated MBI film enhances at 400 nm to 650 nm owing to the good film quality. That can be ascribed to the absorption of MBI, exhibiting an indirect energy bandgap of 2.04 eV [38]. The high absorption of toluene-treated MBI film is speculated to reflect on the photocurrent from the devices comprised of such light absorption layers.

In addition to absorption, the crystallinity of MBI films is also essential for carrier transportation, especially in lead-free material. The minor d orbital coupling effect (spin-orbital coupling effect) in MBI materials, compared to lead-based perovskite materials, gives them a relatively short carrier lifetime [29]. As a result, constructed MBI materials with less defects are of a ladder to perform their intrinsic photovoltaic properties. From a macroscopic point of view, creating high crystallinity materials is speculated to have a minor defect that results in traps for carriers. Based on Figure 6, the characteristic peak of MBI films in XRD patterns indicates that the reactants of methyl ammonium iodide and bismuth iodide completely react and form MBI under both deposition conditions. Diffraction peaks at 26.5°, 33.7°, 37.7°, and 51.5° refer to the characteristic peaks of FTO substrates. Although both of them have no trail of reactants in films, their crystallinity are still different. The diffraction peaks from the toluene-treated MBI film are sharp compared to the peaks from MBI film without treatment. To evaluate the crystallinity of the films, Scherrer equation is used to calculate the averaged grain size of MBI films (Equation (1)).
(1)$$\tau =\frac{0.9\lambda }{\beta cos\left(\theta \right)}$$

Here, τ is the calculated average grain size, λ is the wavelength of the incident X-ray, β is the full width at half maximum (FWHM) of the corresponding diffraction peak. According to Scherrer equation, the average grain size of MBI film without toluene treatment is 24.1 nm, whereas the average grain size of toluene-treated MBI film is 42.1 nm based on the diffraction peak of (101). The enhanced crystallinity of toluene-treated MBI film can be attributed to the improved film formation mechanism with the assistance of anti-solvent dripping during the deposition step. The induced nucleation sites from anti-solvent dripping help MBI film crystalize in metastable conditions. That results in high crystallinity owing to the stable crystallization condition instead of supersaturated condition of the film without toluene treatment [39,40]. The temperature change during the annealing step causes the rapid solubility evolution of different reactants in a wet film. The reactant having low solubility in the solvent system is extracted early and results in a specific element-rich region in the film without toluene treatment.

To reveal element distribution in MBI films, topographies from scanning electron microscope (SEM) and energy-dispersive X-ray spectroscopy (EDS) give information about the effect of anti-solvent on surface morphology and element distribution of MBI films. Figure 7 shows the surface topography and the corresponding element analysis of MBI films with and without anti-solvent treatment. From the top-view image in Figure 7a,g, the MBI film without toluene treatment shows flake-liked precipitation. In contrast, the MBI film with toluene treatment shows smooth surface morphology. The corresponding EDS element mapping of MBI films is demonstrated in Figure 7b,h. To gain insight into the element distribution of MBI films, the bismuth series, iodine series, titanium series, and oxygen series mapping are shown in Figure 7c–f, MBI without toluene treatment, and Figure 7i–l, toluene treated MBI film. The flake-liked region in Figure 7b shows high density of bismuth and the rest of the film show strong titanium and oxygen signal. That indicates poor film coverage and inhomogeneous film formation. In contrast, the morphology from Figure 7g manifests the high film coverage of toluene-treated MBI film. Although a few pinholes can be observed in the topographic image of the MBI film, its homogeneous element distribution in EDS mapping indicates anti-solvent of toluene-assisted deposition can mitigate the aggregation during the spin-coated process. That helps to obtain an MBI film with high film coverage, homogeneous composition, and minor pinhole. The direct contact between the electron transport layer of TiO_2_ and the hole transport layer of PTAA leads to the current leakage and undesired potential shift in the device. Both hinder carrier transport as electron-hole pairs are excited and generated from light striking the device.

Taking advantage of film coverage and minor aggregation, the device with toluene-treated MBI shows high PV performance compared to the device without toluene treatment. The champion device can achieve a PCE of 0.26%, which enhances around 3.2 times, compared with the champion device without toluene treatment of 0.08%, as shown in Figure 8. The smooth *J-V* curve of the champion device with toluene-treated MBI film implies the constructed device can effectively extract the generated electron-hole pairs with the built-in potential in the device.

Table 2 demonstrates the state-of-the-art MBI photovoltaics. The highest PCE of MBI photovoltaics comes from the sequential gas phase deposition, which can achieve to over 3.00%. That can be ascribed to the uniform morphology of deposited MBI film through gas phase reaction. It sheds light on PV performance from morphology manipulation of MBI layers. However, few studies demonstrate the hysteresis effect of MBI photovoltaics even though the device architecture follows n-i-p configuration. In terms of PV performance of a n-i-p device structure, hysteresis effect, describing the mismatch between *J-V* curve from reversed bias and forward bias scan, is a main concern as evaluating PV performance. The hysteresis effect makes PV performance of n-i-p devices hard to be determined. To evaluate the hysteresis effect, the hysteresis index (HI), following Equation (2), can be applied to monitor the degree of hysteresis effect in a device [16]. The calculated HI of the device with toluene-treated MBI is 0.17, whereas the HI of the device without toluene treatment is 0.35 (Figure 8). The reduced HI from the device with toluene-treated MBI is ascribed to its improved film quality.
(2)$$HI=\frac{J_{RC}\left(0.8V_{OC}\right)-J_{FC}\left(0.8V_{OC}\right)}{J_{RC}\left(0.8V_{OC}\right)}$$

## 4. Conclusions

This study successfully demonstrates a lead-free perovskite derivative of methylammonium bismuth iodide as a light absorber layer to construct a non-toxic photovoltaic. A series of anti-solvent is applied to assist the crystallization of methylammonium bismuth iodide films. Among the candidates of anti-solvents, toluene exhibits the lowest dipole moment and the best infiltration ability. Both properties help to form high-quality methyl bismuth iodide film. The improved quality of methyl bismuth iodide films plays a primary role in enhancing the PV performance and mitigating hysteresis phenomena as devices scan from forward or reversed direction. The champion device with toluene-treated MBI methyl bismuth iodide can achieve a PCE of 0.26% with a hysteresis index of 0.169. The results give a clear guideline for preparing a lead-free methyl bismuth iodide film and steps forward to the non-toxic composition of perovskite derivatives.

## Figures and Tables

**Figure 1 nanomaterials-13-00059-f001:**
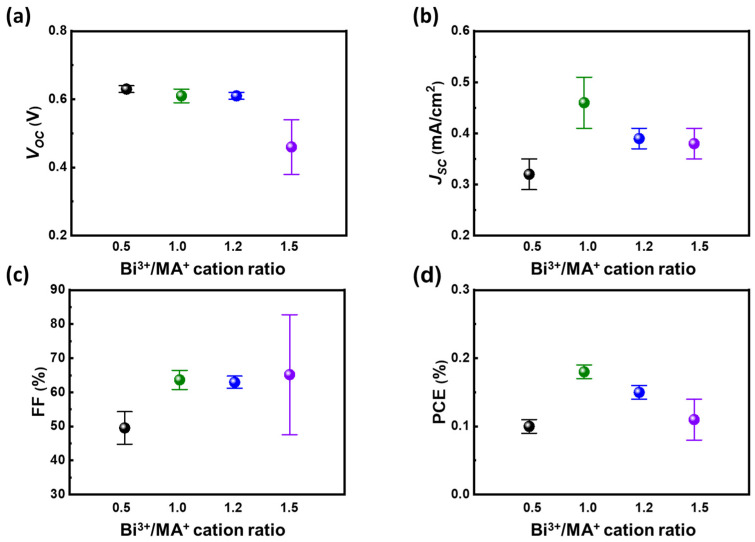
Relation between cation ratio, Bi^3+^/MA^+^ from 0.5 to 1.5, in methylammonium bismuth iodide and their photovoltaic performance: (**a**) *V_OC_*, (**b**) *J_SC_*, (**c**) FF, and (**d**) PCE.

**Figure 2 nanomaterials-13-00059-f002:**
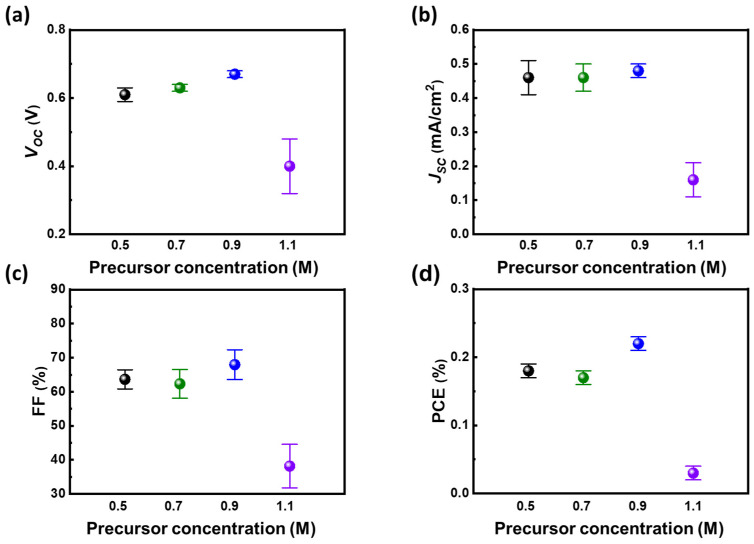
Optimization of MBI materials thickness from precursor concentration: (**a**) *V_OC_*, (**b**) *J_SC_*, (**c**) FF, and (**d**) PCE.

**Figure 3 nanomaterials-13-00059-f003:**
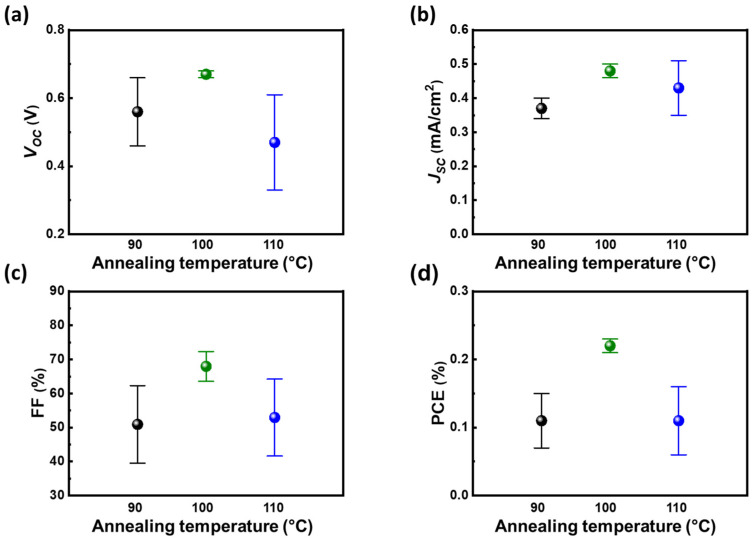
Optimization of annealing temperature, from 90 to 110 °C, for MBI films: (**a**) *V_OC_*, (**b**) *J_SC_*, (**c**) FF, and (**d**) PCE.

**Figure 4 nanomaterials-13-00059-f004:**
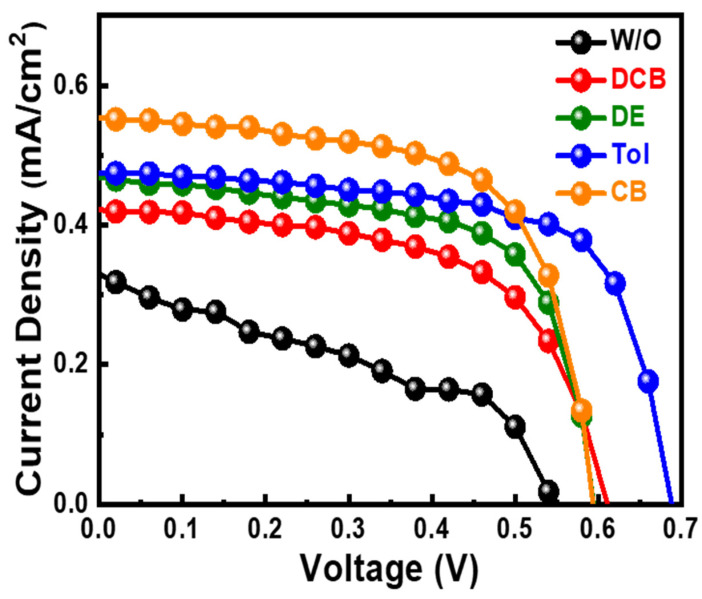
*J-V* curve for solar cells with different antisolvent treated MBI materials.

**Figure 5 nanomaterials-13-00059-f005:**
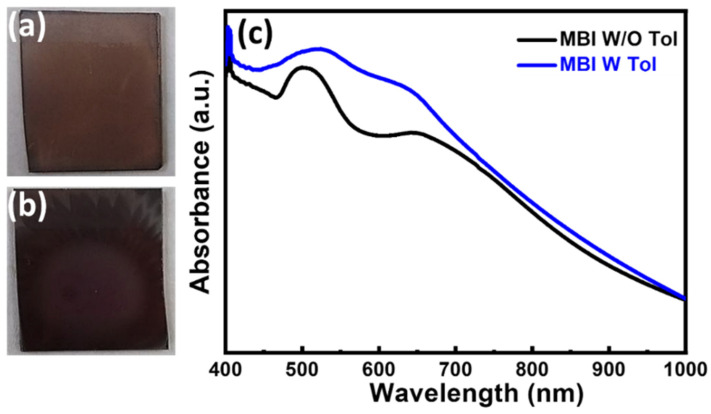
Optical properties of MBI films: (**a**) Appearance of MBI without antisolvent of toluene; (**b**) appearance of MBI with antisolvent of toluene; and (**c**) absorption spectrum of MBI films.

**Figure 6 nanomaterials-13-00059-f006:**
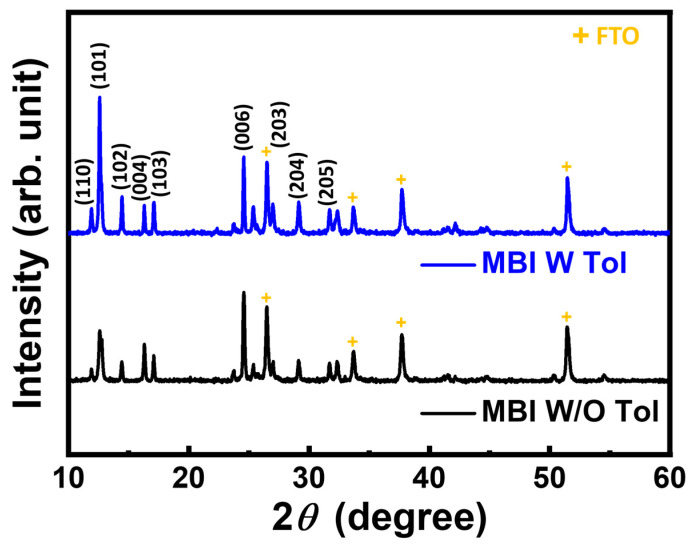
X-ray diffraction pattern of MBI with and without toluene treatment.

**Figure 7 nanomaterials-13-00059-f007:**
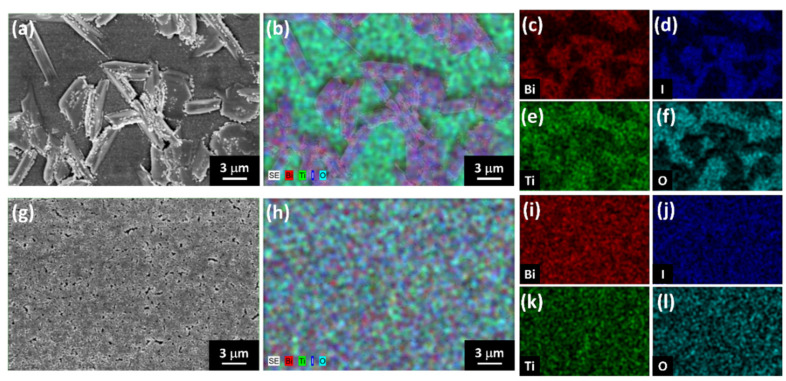
Morphology and element analysis of MBI films without or with anti-solvent of toluene treatment: (**a**) surface morphology of MBI without antisolvent, (**b**) the corresponding EDS, (**c**) bismuth mapping image, (**d**) iodine mapping image, (**e**) titanium mapping image, and (**f**) oxygen mapping image; (**g**) surface morphology of MBI with antisolvent treatment, (**h**) the corresponding EDS, (**i**) bismuth mapping image, (**j**) iodine mapping image, (**k**) titanium mapping image, and (**l**) oxygen mapping image.

**Figure 8 nanomaterials-13-00059-f008:**
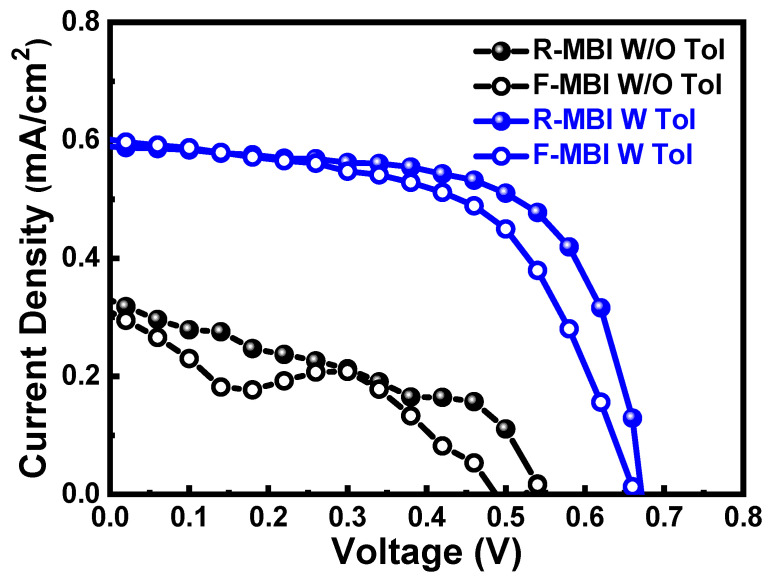
*J-V* curves of champion devices with MBI with or without antisolvent treatment.

**Table 1 nanomaterials-13-00059-t001:** PV performance of devices with MBI films treated with different anti-solvents.

Anti-Solvent	Miscibility	Boiling Point (°C)	Dipole Moment	*V_OC_* (V)	*J_SC_* (mA/cm^2^)	FF (%)	PCE (%)
W/O	-	-	-	0.42 ± 0.19	0.33 ± 0.01	34.98 ± 9.94	0.05 ± 0.03
DE	No	34.6	1.15	0.59 ± 0.02	0.46 ± 0.04	64.72 ± 2.40	0.18 ± 0.02
Tol	No	110.6	0.31	0.67 ± 0.01	0.48 ± 0.02	67.96 ± 4.38	0.22 ± 0.01
CB	Yes	132.0	1.54	0.60 ± 0.01	0.56 ± 0.03	63.73 ± 2.38	0.21 ± 0.01
DCB	Yes	180.2	2.70	0.60 ± 0.03	0.41 ± 0.02	60.83 ± 3.69	0.15 ± 0.01

**Table 2 nanomaterials-13-00059-t002:** State-of-the-art PV performance of MBI photovoltaics (PCE > 0.1%).

Device Structure	Processing	*V_OC_* (mV)	*J_SC_* (mA/cm^2^)	*FF* (%)	PCE (%)	Ref.
FTO/c-TiO_2_/meso-TiO_2_/MA_3_Bi_2_I_9_ /P3HT/Au	Gas phase deposition	1.01	4.02	78	3.17	[41]
FTO/c-TiO_2_/mp-TiO_2_/MA_3_Bi_2_I_9_ /spiro-OMeTAD/Au	Gas phase deposition	0.81	2.95	69	1.64	[30]
FTO/PEDOT:PSS/MA_3_Bi_2_I_9_/C60/BCP/Ag	Gas phase deposition	0.83	1.39	34	0.39	[29]
FTO/c-TiO_2_/mp-TiO_2_/MA_3_Bi_2_I_9_ /spiro-OMeTAD/Au	Gas phase deposition	0.59	1.18	48	0.33	[42]
FTO/Ti/m-TiO_2_/MA_3_Bi_2_I_9_/spiro-MeOTAD/Au	Gas phase deposition	0.595	0.695	41	0.17	[43]
FTO/c-TiO_2_/mp-TiO_2_/MA_3_Bi_2_I_9_ /spiro-OMeTAD/Au	Gas phase deposition	0.72	0.61	38	0.17	[44]
FTO/c-TiO_2_/mp-TiO_2_/MA_2_Bi_2_I_9_/P3HT/carbon	Solution deposition	0.87	2.70	69	1.62	[26]
FTO/c-TiO_2_/m-TiO_2_/MA_3_Bi_2_I_9_ NPs/carbon	Solution deposition	0.62	1.86	56	0.65	[45]
FTO/TiO_2_/MA_3_Bi_2_I_9_/spiro-OMeTAD/Au	Solution deposition	0.626	1.12	48	0.34	[46]
FTO/mp-TiO_2_/MA_3_Bi_2_I_9_ /spiro-OMeTAD/Au	Solution deposition	0.61	1.12	43	0.29	[47]
FTO/c-TiO_2_/meso-TiO_2_/MA_3_Bi_2_I_9_ /spiro-OMeTAD/Au	Solution deposition	0.56	0.83	48	0.26	[48]
FTO/c-TiO_2_/mp-TiO_2_/MA_3_Bi_2_I_9_ /PTAA/Ag	Solution deposition	0.59	1.07	42	0.26	[49]
FTO/bl-TiO_2_/m-TiO_2_/MA_3_Bi_2_I_9_/P3HT/Au	Solution deposition	0.354	1.16	46	0.19	[50]
FTO/Ti/c-TiO_2_/MA_3_Bi_2_I_9_ /spiro-OMeTAD/Au	Solution deposition	0.72	0.61	38	0.17	[51]
FTO/c-TiO_2_/mp-TiO_2_/MA_3_Bi_2_I_9_ /spiro-OMeTAD/Au	Solution deposition	0.21	2.33	33	0.17	[52]
FTO/c-TiO_2_/mp-TiO_2_/MA_3_Bi_2_I_9_ /spiro-OMeTAD/Au/Ag	Solution deposition	0.59	0.50	57	0.17	[53]
FTO/c-TiO_2_/meso-TiO_2_/MA_3_Bi_2_I_9_ /spiro-MeOTAD/Ag	Solution deposition	0.68	0.52	33	0.12	[54]

## Data Availability

Data are available from the corresponding author upon reasonable request.

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
