# Peer review of "Antisolvent Engineering to Enhance Photovoltaic Performance of Methylammonium Bismuth Iodide Solar Cells"

_nanomaterials, 2022, doi:10.3390/nano13010059_

Round 1
Reviewer 1 Report
Wu et al. report on the production of model photovoltaic cells based on methylammonium bismuth iodide. The authors attribute the importance of such materials to the need to dispense with lead-based photovoltaic cells. A bismuth photovoltaic layer can be deposited on a silica substrate either directly from the mother liquor or by adding non-polar organic solvents to the mother liquor. The use of these organic solvents makes it possible to obtain layers with significantly higher energy conversion efficiencies. To explain the reasons for this increase, a qualitative comparison of the morphology of methylammonium bismuth iodide films on the surface of a quartz substrate was performed. The comparison showed an increase in the crystallinity of the films when an organic solvent was added.
The most significant disadvantage of the work is that the resulting photovoltaic layers have a very low energy conversion efficiency. Even after increasing this efficiency by using an organic solvent, it remains one to two orders of magnitude lower than that of the best photovoltaic cells. The authors should have been aware of this problem before starting the study, but they do not go into it in any way in the introduction.
In general, the article was written very carelessly. Many abbreviations are not explained. Why the morphology of films on quartz and amorphous silica should be comparable is unclear.
The conclusions of the article often do not follow from the results of the work, and sometimes they are not discussed in the work at all.
Abstract:
“The spin-orbital coupling effect of lead-based perovskite materials makes them exhibit a long carrier lifetime and plays an essential role in their photovoltaic performance.”
The spin-orbital coupling effect is not discussed in the article.
“Owing to the short carrier diffusion length of MBI, its film quality is a predominate factor to photovoltaic performance.”
The carrier diffusion length of MBI was not studied in the article.
"The improved morphology and crystallinity of MBI films promote photovoltaic performance over 2.5 times compared with the one without toluene treatment. The photovoltaic device can achieve 0.26% …”
Table 1 provides other numbers.
4. Conclusions
"This study successfully demonstrates a lead-free perovskite-derivatives of methylammonium bismuth iodide as a light absorber layer to construct a non-toxic photovoltaic."
The resulting photovoltaic layers have a very low energy conversion efficiency.
"The improved quality of methyl bismuth iodide films plays a primary role in enhancing the PV performance and mitigating hysteresis phenomena as devices scan from forward or reversed direction.”
These conclusions are not justified.
Author Response
Please see the attachment. The revised paragraphs and additional information in revised manuscript are highlighted in yellow.

Reviewer 2 Report
The authors presented an interesting work about the antisolvent engineering to enhance photovoltaic performance of MBI.
The paper is well structured but some information about the state of art and the comparison of the results with the current status of the art.
As an example, in the case of the figure 1C, the ratio of Bi3+/MA+ 1.5 presents a huge variation. Can you, please, provide some reasons?
Please, explain the obtained results and provide some reasons about the results obtained, compared with the literature.
Author Response
Please see the attachment. The revised paragraphs in revised manuscript are highlighted in yellow.

Round 2
Reviewer 1 Report
Although the overall level of the article has remained very low, its revised version may be of interest to readers, mainly due to Table 2.
Reviewer 2 Report
Thank you for addressing the changes.
Regards.